# Mixed Depression in the Post-COVID-19 Syndrome: Correlation between Excitatory Symptoms in Depression and Physical Burden after COVID-19

**DOI:** 10.3390/brainsci13040688

**Published:** 2023-04-20

**Authors:** Alessio Simonetti, Evelina Bernardi, Stella Margoni, Antonello Catinari, Antonio Restaino, Valentina Ieritano, Marta Palazzetti, Federico Mastrantonio, Delfina Janiri, Matteo Tosato, Francesco Landi, Gabriele Sani

**Affiliations:** 1Department of Neuroscience, Section of Psychiatry, Fondazione Policlinico Universitario Agostino Gemelli IRCCS, 00168 Rome, Italygabriele.sani@unicatt.it (G.S.); 2Menninger Department of Psychiatry and Behavioral Sciences, Baylor College of Medicine, Houston, TX 77030, USA; 3Department of Geriatrics, Università Cattolica del Sacro Cuore, 00168 Rome, Italy; 4Department of Geriatrics, Fondazione Policlinico Universitario Agostino Gemelli IRCCS, 00168 Rome, Italy; 5Department of Neurosciences, Section of Psychiatry, Università Cattolica del Sacro Cuore, 00168 Rome, Italy

**Keywords:** mixed depression, depression, agitation, COVID-19, post-COVID-19 syndrome

## Abstract

The relationship between depression and post-COVID-19 disease syndrome (post-COVID-19 syndrome) is established. Nevertheless, few studies have investigated the association between post-COVID-19 syndrome and mixed depression, i.e., a specific sub-form of depression characterized by high level of excitatory symptoms. Aims of the present study are: (a) to compare the post-COVID-19 syndrome’s burden in depressed and non-depressed patients, and (b) to investigate the correlation between post-COVID-19 syndrome’s burden and the severity of mixed depression. One thousand and forty six (n = 1460) subjects with post-COVID-19 syndrome were assessed. Subjects were divided into those with (DEP) or without (CONT) depression. Sociodemographically, post-COVID-19 syndrome’s symptoms number and type were compared. In DEP, association between levels of excitatory symptoms and the presence of post-COVID-19 syndrome’s symptoms were additionally assessed. DEP showed greater percentages of family history of psychiatric disorders than CONT. DEP showed higher percentages of post-COVID-19 symptoms than CONT. A greater level of excitatory symptoms were associated to higher frequencies of post-COVID-19 syndrome’ symptoms. Higher levels of post-COVID-19 syndrome’s symptoms in DEP corroborate the evidence of a common pathway between these two syndromes. Presence of excitatory symptoms seem to additionally add a greater illness burden. Such findings might help clinicians choose the appropriate treatment for such states. More specifically, therapies aimed to treat excitatory symptoms, such as antipsychotics and mood stabilizers, might help reduce the illness burden in post-COVID-19 patients with mixed depression.

## 1. Introduction

From an initial cluster of pneumonia cases reported in Wuhan (December 2019) [1], the SARS-CoV-2 infection has spread globally, officially becoming a pandemic in March 2020 [2,3]. SARS-CoV-2 infection displays great clinical variability, as many patients may remain asymptomatic while others may develop severe forms of interstitial pneumonia, acute respiratory distress syndrome (ARDS), and multi-organ failure (MOF).

A growing group of cases also reported a set of symptoms persisting for weeks or months after the acute phase of illness. The clinical sequelae of SARS-CoV-2 infection are called post-COVID-19 syndrome or long-COVID-19 syndrome. This syndrome is defined by persistent clinical signs and symptoms that appear after COVID-19 disease, persist for more than 12 weeks, and cannot be accounted for by any alternative diagnosis. These symptoms include cough, dyspnoea, fatigue, difficulties with memory and concentration, sleep disorders, gastrointestinal complaints, and musculoskeletal problems [4,5]. Psychiatric syndromes, including depression, anxiety, sleep disturbances, and post-traumatic disorder, have also been reported [6,7,8].

Among the psychiatric symptoms related to the COVID-19 pandemic, depression represents one of the major public health concerns worldwide. Rates of depressive symptoms persisting after SARS-CoV-2 infection have risen up to 28%, and depressive symptoms during post-COVID-19 syndrome are associated with greater functional impairment and a general worsening of the quality of life [9]. Mechanisms linking depression and SARS-CoV-2 infection sequelae are not known. Nevertheless, the extant literature suggests a common pathogenesis developing through serotonin imbalance, hypothalamic–pituitary–adrenal (HPA) axis dysfunction, neuroplastic downplay, and disruption of affective circuits [10].

The available studies focused mainly on the association of depressive symptoms with other neuropsychiatric symptoms, such as anxiety, delirium, post-traumatic stress disorder, insomnia, and obsessive–compulsive symptoms [10,11,12,13,14], whereas data on the association between depression and other typical post-COVID-19 syndrome’s symptoms are scarce.

Bucciarelli et al. [15] hypothesized an association between depression after SARS-CoV-2 infection and cardiovascular risk, mainly because of reduced physical activity and deteriorating lifestyle habits. Mazza et al. [10,16] also found a high level of fatigue and pain in COVID-19 survivors with depression. Post-COVID-19 syndrome’s associated depression has been also related to microbiome alteration [17] and higher incidence of gastrointestinal symptoms such as heartburn, constipation, diarrhea, and abdominal pain [18]. However, the reliability of the aforementioned studies is hampered by small sample sizes and the narrow range of post-COVID-19 syndrome’s symptoms that have been assessed.

Further uncertainty comes from the fact that the aforementioned literature lacks an assessment of several subforms of depression, which embed distinct psychopathology, phenotypes, and courses. In recent years, greater attention has been given to a specific form of depression named “mixed depression” [19,20,21]. Mixed depression endorses core features of depression, i.e., depressed mood, anhedonia, and lack of interest with excitatory symptoms, namely, psychomotor agitation, inner tension, racing or crowded thoughts, early insomnia, and lack of retardation [20,21]. Psychomotor agitation is present in many cases, but not in all. Depression, an anxious mood and inner psychic agitation dominate the clinical picture of mixed depression. In the cases without psychomotor agitation, inner unrest is the main symptom. This inner agitation makes the patient very anxious and fearful [22]. There is typically a disturbance of the train of thought called crowded or racing thoughts. Lability of mood and emotional reactivity is also characteristic of the clinical picture of mixed depression. Because of the characteristic great energy and impulsivity of mixed depression, the risk of suicide is very high. Compared with nonmixed depression, mixed depression is held to be more severe and more common in bipolar disorders. In women, it is associated with BD-II and a hyperthymic temperament [23]. Mixed depression is also associated with younger age at onset, more family history of BD, and poorer response to treatments [23,24,25,26,27,28]. Mixed depression is characterized by more severe depressive symptoms, worse outcome, lower family quality of life, poorer global functioning, more externalizing problems, increased presence of comorbidities, and greater suicide risk than non-mixed depression [19,20,21,22,23,24,25,26,27,28,29,30,31]. Such greater illness burden might be driven by its excitatory compounds [32], which may be subsided by a more dysregulated neurobiological substrate [21].

Therefore, despite the increasing number of studies associating depression with long-COVID syndrome, there is a need to corroborate the already existing data with larger samples sizes and a broader array of post-COVID-19 syndrome’s symptoms. Furthermore, there are no studies investigating the effects of mixed symptoms in patients with depression after COVID-19.

### Aims of the Study

The aim of this study was to investigate the relationship between depression and a large number of post-COVID-19 syndrome’s symptoms. Furthermore, the aim of this study is to investigate the relationship between frequency and severity of post-COVID-19 syndrome’s symptoms in patients with depression and mixed symptoms.

We expected that patients with depression would show more post-COVID-19 syndrome symptoms than those without depression. Since mixed symptoms are associated with a worse illness burden and have been associated with greater neurobiological imbalance than those with non-mixed depression, we expected mixed symptoms to correlate with a greater presence of post COVID-19 syndrome’s symptoms.

## 2. Materials and Methods

### 2.1. Sample

The sample consisted of a cohort of 1460 subjects who suffered from SARS-CoV-2 infection. Subjects underwent a multidisciplinary evaluation in the Post-Acute Care Service at the Fondazione Policlinico Universitario Agostino Gemelli IRCCS of Rome, Italy (Gemelli Against COVID-19 Post-Acute Care Service) from 21 April 2020 to 11 May 2022. The assessment made included: (i) collection of detailed medical history; (ii) physical examination; and (iii) internal medicine, geriatric, ophthalmological, otolaryngologic, pneumological, psychiatric, cardiological, immunological, and rheumatological evaluations. The following inclusion criteria were applied: (a) age between 18 and 75; (b) previous positivity to COVID-19; and (c) capability of providing informed consent. Exclusion criteria were: severe neurodevelopmental disorders, dementia, or other severe neurological disorders, and the presence of depression prior to COVID-19 infection. Subjects involved gave their written informed consent to the study. The study was approved by the Ethical Committee of the Fondazione Policlinico Universitario Agostino Gemelli IRCCS (protocol number: 0013008/20). Variables considered for the present study were: (a) sociodemographic characteristics (i.e., age, gender, presence/absence of employment, and education), and (b) data regarding psychiatric evaluation in the Post-Acute Care Service at the Fondazione Policlinico Universitario Agostino Gemelli IRCCS of Rome, and, more specifically, data regarding psychiatric evaluation included: presence/absence of psychiatric history, frequency of past psychotropic drugs assumed, frequency of past psychotherapy, presence/absence of psychiatric history in first-degree relatives, and evaluation of current psychopathology through rating scales; and (c) data regarding the aforementioned multidisciplinary assessment in the Post-Acute Care Service at the Fondazione Policlinico Universitario Agostino Gemelli IRCCS of Rome.

Primary outcomes of the study were: (a) to compare the post-COVID-19 syndrome’s burden in DEP and CONT, and (b) to assess the correlation between post-COVID-19 syndrome’s burden and the severity of mixed depression.

### 2.2. Assessment

The sample was divided into two groups according to the presence/absence of depressive symptoms. Subject of (1) with an HAM-D total score > 7 were defined as subjects with depression (DEP). Subjects with a HAM-D score < 7 were defined as comparisons (COMP). The cut-off of 7 was chosen according to the vast majority of the antidepressant’s clinical trials, which set a score of HAM-D < 7 as the cut-off point to define absence/remission of depression [33]. COMP also needed to have an absence of unstabilized psychopathology, as confirmed by a Hamilton Anxiety Rating Scale (HAM-A) score below 17 and a Brief Psychiatric Rating Scale (BPRS) total score below 25.

Detailed description of psychiatric rating scales used were provided below:

The HAM-D [34] is used to assess severity of depressive symptoms. Scores of 0–7 are indicative of the absence of depression; scores of 8–16 suggest mild depression; scores of 17–23 are indicative of moderate depression; and scores over 24 are indicative of severe depression.

The HAM-A [35] is a questionnaire used to measure the severity of anxiety symptoms. Total scores range between 0 and 56. Scores < 17 indicate mild anxiety, 18–24 mild to moderate anxiety, 25–30 moderate to severe anxiety, and >30 severe anxiety.

The BPRS [36] was developed as a measurement general psychopathology. Possible scores vary from 24 to 168, with lower scores indicating less severe psychopathology.

The Koukopoulos Mixed Depression Rating Scale (KMDRS) [37] is a self-administered rating scale, consisting of 14 items evaluating the presence and severity of the typical symptoms of mixed depression. Possible scores range from a minimum of 0 to a maximum of 51, with higher scores indicating greater severity of mixed depressive symptoms.

### 2.3. Statistical Analysis

Descriptive analyses of the sample were initially investigated.

Between-group differences were analyzed with the chi-squared test (χ^2^) for nominal variables and *t*-tests for continuous variables. In each of the *t*/χ^2^-tests, groups (DEP, CONT) were independent variables, while sociodemographic characteristics (i.e., age, gender, presence/absence of employment, education), data regarding the post-COVID-19 syndrome (i.e., time elapsed from COVID-19, type and number of symptoms present during COVID-19, presence/absence of hospitalization due to COVID-19, hospitalization length, number and type of symptoms during the evaluation, such as that during post-COVID-19 syndrome), number and types of current medications, and, finally, data regarding psychiatric evaluation (presence/absence of psychiatric history, presence/absence of psychiatric history in the first-degree relatives, number of past psychotropic drugs, presence/absence of previous psychotherapy, number of psychotropic drugs at the time of evaluation, psychopathological scales total scores) were all dependent variables.

The relationship between mixed symptoms and the presence of post-COVID-19 symptoms in DEP was investigated with binary logistic regression. In each regression, KMDRS scores were independent variables and the presence/absence of post-COVID19 syndrome’s symptoms were dependent variables. Additionally, the relationship between KMDRS scores and the frequency of overall, psychiatric, cardiac, pneumological, endocrinological, rheumatological, and medication assumed have all been investigated. We used the statistical routines of SPSS Statistics 24.0 for Windows (IBMCo., Armonk, New York, NY, USA, 2016).

## 3. Results

### 3.1. Main Characteristics of the Sample

Sociodemographic characteristics and data regarding psychiatric evaluation are presented in Table 1. A total of 385 subjects (26% of the whole sample) were classified as DEP. DEP showed higher rates of females, and a higher percentages of subjects with psychiatric history than CONT. DEP also more frequently underwent psychopharmacological treatments and psychotherapy, and showed a higher percentage of family history for psychiatric disorders than CONT.

### 3.2. Differences in COVID-19 Related Characteristics and Post-COVID-19 Syndrome’s Symptoms

DEP showed greater levels of hospitalization for COVID-19, and assumed more frequently cardiac and psychiatric medications than CONT. DEP showed higher rates of fatigue, cough, diarrhea, headache, anosmia, dysgeusia, red eyes, low vision, syncope, vertigo, joint pain, Sjogren’s syndrome, myalgia, dyspnea, chest pain, sore throat, rhinitis, and lack of appetite than CONT. Comparisons made and significant differences are presented in Table 2.

### 3.3. Relationship between Severity of Mixed Symptoms and Post-COVID-19 Syndrome’s Symptoms

Regressions made in DEP showed that higher levels of KMDRS total scores were associated with greater percentages of diarrhea, fatigue, headache, vertigo, joint pain, myalgia, dyspnea, and chest pain.

### 3.4. Effect of Possible Confounding Variables

All the regressions were corrected for the effect of possible confounding variables, i.e., age, presence/absence of hospital admission due to COVID-19, number of symptoms during SARS-CoV-2 infection, and time elapsed from COVID-19. Regression coefficients and *p* values are shown in Table 3. No effect of the aforementioned variables on the results that were found emerged.

## 4. Discussion

Results emerging from our study showed that patients with depression and post-COVID-19 syndrome have higher proportions of psychiatric diagnoses in first-degree relatives, higher percentages of subjects who underwent psychopharmacological treatment, and higher rates of fatigue, cough, diarrhea, headache, anosmia, dysgeusia, red eyes, low vision, syncope, vertigo, joint pain, Sjogren’s syndrome, myalgia, dyspnea, chest pain, sore throat, rhinitis, and lack of appetite than people without depression. Regressions made showed that higher levels of excitatory symptoms in depressed patients are associated with a greater incidence of diarrhea, fatigue, headache, vertigo, joint pain, myalgia, dyspnea, and chest pain.

Our results support the available literature, documenting higher rates of depression in the post-COVID-19 syndrome as compared to the general population [38,39], and are in line with those documenting a high level of psychopathology in depressed subjects after SARS-CoV-2 infection [39]. Furthermore, our study reported that the familial transmission observed in the affective disorder also applies to the subgroup of those affected by the post-COVID-19 syndrome. Regarding physical symptoms, our results are in line with those documenting a relationship between depression and fatigue and pain in COVID-19 survivors [10,16], and those documenting alterations in microbiome [17]. In addition to these studies, we reported higher rates of cough, headache, anosmia, dysgeusia, red eyes, low vision, syncope, vertigo, Sjogren’s syndrome, dyspnea, chest pain, sore throat, rhinitis, and lack of appetite in subjects with depression. The etiology of the association found is not known, and yet, nonetheless, shared neuroinflammatory mechanisms might account for the relationship between COVID-19 sequelae and depression. SARS-CoV-2 infection can induce cytokine dysregulation by activating the mounting release of pro-inflammatory cytokines (e.g., IL-1β, IL-6, and tumor necrosis factor-alpha) and the inhibition of anti-inflammatory type-I interferon responses, thus causing “the cytokine storm”. Cytokine dysregulation drives systemic peripheral inflammatory responses [40,41] that have been shown to take part to the pathophysiology of the symptoms that studies associate with depression, i.e., cough [42], headache [43], anosmia [44], dysgeusia [45], red eyes, low vision [46], syncope [47], vertigo [48], Sjogren’s syndrome [49], dyspnea, chest pain [50], sore throat [51], rhinitis [52], and lack of appetite [53]. At the same time, cytokine dysregulation can compromise the function of the blood-brain barrier (BBB), leading to pro-inflammatory cytokines passing the leaky BBB and activating microglia. Chronic microglial activation enhances the synthesis and release of pro-inflammatory cytokines, driving neuro-inflammation, and, possibly, participates in the development of depressive symptoms [38]. We found that mixed symptoms in subjects with depression correlate with higher percentages of diarrhea, fatigue, headache, vertigo, joint pain, myalgia, dyspnea, and chest pain. This relationship might suggest that the greater severity of excitatory symptoms in depression are associated with the greater severity of the post-COVID-19 syndrome. Mixed depression embeds an array of severe neurobiological alterations, such as hyperinflammation and the imbalance of monoamines [21]. More specifically, mixed depression has been related to a higher level of dopamine, norepinephrine, and serotonin than non-mixed depression [54,55,56]. Mixed depression is also associated with dexamethasone–suppression test non-suppression [32] and cortisol level fluctuation [57], suggesting a hypothalamic-pituitary-adrenal (HPA axis) imbalance. These mechanisms have also been related to symptoms experienced after COVID-19 infection. High blood serotonin levels were associated to a high prevalence of diarrhea-predominant irritable bowel syndrome [58], whereas injection of serotonin precursor L-5-hydroxytryptophan (L-5-HTP) in mice has been associated with a steep increment of defecation [59]. Attenuation of the HPA axis and enhancement of the sympathetic/adrenal medulla (SAM) system have been related to the onset and maintenance of fatigue [60,61]. Headache has been associated with hyperinflammation via persistent immune system and trigemino-vascular activation [62]. Noradrenaline application to cultured dural afferents also increased action potential firing in response to the current application, contributing to pro-nociceptive signaling from the meninges via actions on dural afferents and dural fibroblasts [63]. This suggests a monoamine-mediated onset of headaches. Aberrant monoamine release might also explain the relationship between mixed symptoms and rates of myalgia, joint pain, and chest pain. Serotonin peripheral stimulation has been related to nociception [64,65], and descending noradrenergic and serotonergic pathways have been shown to inhibit nociception coming from musculoskeletal system [66]. Aberrant serotonin and noradrenalin transmission have been proven to alter these pathways, leading to a heightened sensitivity to pain and even a painful reaction to normally non-noxious stimuli [67]. Additionally, aberrant serotonin transmission appears to maintain chronic myalgia through modulation of local muscle microcirculation [68]. Relationships between mixed symptoms and vertigo might involve multiple common underlying mechanisms. To this extent, altered HPA axis might trigger the pro-inflammatory M1 phenotype microglia. Such activation might result in massive cytokines, nitrogen, and oxygen free radical (RNS/ROS) release [68,69] that can alter vestibular homeostasis and cause vertigo, sweat, and nausea. Additionally, Pompeiano et al. [70] and Yates et al. [71] found that altered serotonergic and noradrenergic firing from the raphe nuclei and locus coeruleus might induce vertigo through altered vestibulo-spinal reflex. Finally, aberrant release of dopamine and serotonin from the substantia nigra and the ventral tegmental area have proven to induce hyperventilation [72] and insufficient skeletal muscle energy status and autonomic dysfunction [73]. This may explain the relationship between the higher level of mixed symptoms and the greater occurrence of dyspnea.

### 4.1. Strenghts and Limitations of the Study

Several limitations should be mentioned. The cross-sectional nature of the present work impedes the clearly identification of the relationship between post-COVID-19 syndrome, psychopathology, and mixed depression. Therefore, hypothesis made on the possible relationship between mixed depression and physical symptoms during post-COVID-19 syndrome are speculative. Accordingly, since we were unable to evaluate putative alterations underlying either the post-COVID-19 syndrome and mixed symptoms, mechanisms involved in the relationship between psychopathology and physical symptoms are speculative in nature. In the present work, we did not include instrumental analyses in our evaluation. Spirometry, the six-minute walking test, and a comprehensive cognitive battery could have added or precise measurements of impairments related to mixed states. Finally, the present study’s information on the psychiatric diagnosis and psychotropic drugs assumed is limited. Since such variables have proven to affect brain morphology and behavior [21,74,75,76,77,78], further studies are needed to evaluate the effect of psychiatric diagnosis and psychotropic drugs in subjects with post-COVID-19 syndrome.

On the other hand, the present study’s large sample size represents a strength of the study. Furthermore, the availability of performing comprehensive and multidisciplinary evaluations adds further knowledge to the number and severity of multiorgan alterations related to the post-COVID-19 syndrome and their relationship with psychopathology.

### 4.2. Future Perspectives

The correlation found between post-COVID-19 symptoms and mixed depression could be linked to a common dysregulation of immune and neurotransmitter systems, which might also be related to the neuroinflammatory effect of the infection.

The evidence of a common pathway between post-COVID-19 syndrome and mixed depression suggests that treatment strategies aimed to reduce the severity of mixed symptoms might also lower post-COVID-19 syndrome’s burden. Mixed depression benefits from the effect of anti-excitatory drugs such as antipsychotics and mood stabilizers, whereas it might worsen with antidepressants [23,77]. Because of the relationship between mixed depression and post-COVID-19 syndrome’s burden, treatment with antidepressants and mood stabilizers might improve both the physical condition and mood in those suffering from COVID-19 sequelae.

## 5. Conclusions

The present findings corroborate the close relationship between post-COVID-19 syndrome and depression. Furthermore, the present findings highlight that excitatory symptoms in depression are associated with a greater illness burden. Even though the etiology of this relationship is not known, immune system dysregulation and monoamine imbalance might be the link between hyperarousal and psychomotor agitation and the physical symptoms experienced after COVID-19. Further studies are warranted to corroborate the present findings and to tailor specific treatment for reducing excitatory symptoms and illness burden after SARS-CoV-2 infection.

## Figures and Tables

**Table 1 brainsci-13-00688-t001:** Sociodemographic characteristics and data regarding psychiatric evaluation.

Variables	DEP	CONT	*t*/χ	*p*
**Age, mean ± SD**	**53.70 ± 13.34**	**56.35 ± 14.53**	**9.774**	**0.002**
**Females, n (%)**	**256 (66.5)**	**418 (38.9)**	**86.692**	**0.000**
Education, mean ± SD	13.16 ± 5.154	13.36 ± 5.192	0.436	0.509
Employed, n (%)	265 (68.8)	693 (64.5)	2.331	0.127
**Psychiatric hystory**	**78 (20.3)**	**55 (5.1)**	**78.397**	**0.000**
**Previous psychopharmacotherapy**	**48 (12.5)**	**31 (2.9)**	**50.798**	**0.000**
**Previous psychotherapy**	**47 (12.2)**	**46 (4.3)**	**29.823**	**0.000**
Previous use of substances	66 (17.1)	181 (16.9)	0.017	0.896
**Psychiatric history in relatives**	**53 (13.8)**	**71 (6.6)**	**18.659**	**0.000**
**HAM-A, mean ± SD**	**12.28 ± 5.63**	**2.09 ± 2.15**	**1.357**	**0.000**
**BPRS, mean ± SD**	**30.35 ± 5.11**	**24.86 ± 3.19**	**326.085**	**0.000**

Note: Significant results are in bold. CONT: subjects without depressive symptoms; DEP: subjects with depressive symptoms. BPRS: Brief Psychiatric Rating Scale; HAM-A: Hamilton Anxiety Rating Scale.

**Table 2 brainsci-13-00688-t002:** Differences in COVID-19 related characteristics and post-COVID-19 syndrome’s symptoms.

Variables	DEP (N, %)	CONT (N, %)	*t*/χ	*p*
**Characteristics related to COVID-19**				
**Hospitalization**	**201, 52.2%**	**639, 59.5%**	**6.165**	**0.013**
Current medications	269, 69.9%	718, 66.9%	1.179	0.278
**Current cardiac medications**	**16, 4.5%**	**83, 8.3%**	**5.296**	**0.021**
Current pneumological medications	6, 1.7%	10, 1.0%	1.131	0.288
Current endocrinological medications	18, 5.1%	53, 5.3%	0.013	0.911
Current rheumatological medications	3, 0.9%	8, 0.8%	0.011	0.918
**Current psychiatric medications**	**59, 15.3%**	**61, 5.7%**	**34.93**	**0.000**
Other current medications	35, 9.9%	89, 8.8%	0.378	0.539
Current polytherapy	158, 44.9%	407, 40.5%	2.105	0.147
**Post COVID-19 syndrome’s symptoms**				
Fever	10, 2.6%	13, 1.2%	3.514	0.061
**Fatigue**	**291, 75.6%**	**588, 54.7%**	**51.372**	**0.000**
**Cough**	**75, 19.5%**	**125, 11.6%**	**14.733**	**0.000**
**Diarrhea**	**50, 13.0%**	**75, 7.0%**	**13.041**	**0.000**
**Headache**	**128, 33.2%**	**172, 16.0%**	**51.520**	**0.000**
**Anosmia**	**85, 22.1%**	**142, 13.2%**	**16.920**	**0.000**
**Dysgeusia**	**79, 20.5%**	**114, 10.6%**	**24.223**	**0.000**
**Red eyes**	**41, 10.6%**	**64, 6.0%**	**9.335**	**0.002**
**Low vision**	**101, 26.2%**	**188, 17.5%**	**13.595**	**0.000**
**Syncope**	**6, 1.6%**	**5, 0.5%**	**4.524**	**0.033**
**Vertigo**	**73, 19.0%**	**119, 11.1%**	**15.403**	**0.000**
**Joint pain**	**172, 44.7%**	**313, 29.1%**	**30.809**	**0.000**
Skin lesion	35, 9.1%	72, 6.7%	2.376	0.123
**Sjogren’s syndrome**	**72, 18.7%**	**106, 9.9%**	**20.636**	**0.000**
Raynaud phenomenon	8, 2.1%	14, 1.3%	1.144	0.285
**Myalgia**	**170, 44.2%**	**285, 26.5%**	**40.998**	**0.000**
**Dyspnea**	**267, 69.4%**	**625, 58.2%**	**14.848**	**0.000**
**Chest pain**	**123, 31.9%**	**169, 15.7%**	**46.533**	**0.000**
**Sore throat**	**36, 9.4%**	**47, 4.4%**	**13.071**	**0.000**
Sputum	27, 7.0%	68, 6.3%	0.216	0.642
**Rhinitis**	**42, 10.9%**	**78, 7.3%**	**4.992**	**0.025**
**Lack of appetite**	**37, 9.6%**	**54, 5.0%**	**10.176**	**0.001**

Significant results are in bold. CONT: subjects without depressive symptoms; DEP: subjects with depressive symptoms.

**Table 3 brainsci-13-00688-t003:** Relationship between severity of mixed symptoms and current medications and post-COVID-19 syndrome’s symptoms in DEP.

Variable	F	*p*	Wald	OR	CI (Lower-Upper)
Medications	0.001	0.959	0.003	1.001	0.949 1.057
**Psychiatric medications**	**0.140**	**0.000**	**15.321**	**1.150**	**1.072 1.234**
Cardiac medications	−0.125	0.157	2.000	0.882	0.741 1.050
Pneumological medications	−0.053	0.647	0.210	0.948	0.756 1.189
Endocrinological medications	0.092	0.065	3.414	1.097	0.994 1.209
Rheumatological medications	−0.017	0.910	0.013	0.983	0.737 1.312
Other medications	0.028	0.492	0.473	1.029	0.949 1.115
Polytherapy	−0.004	0.893	0.018	0.997	0.947 1.049
Fever	0.079	0.241	1.372	1.082	0.948 1.234
Fatigue	0.018	0.560	0.340	1.018	0.959 1.080
Cough	−0.010	0.765	0.090	0.990	0.930 1.055
**Diarrhea**	**0.089**	**0.007**	**7.162**	**1.093**	**1.024 1.167**
**Headache**	**0.082**	**0.002**	**9.828**	**1.086**	**1.031 1.143**
Anosmia	−0.015	0.640	0.219	0.986	0.927 1.047
Dysgeusia	0.022	0.462	0.541	1.023	0.964 1.085
Red eyes	0.029	0.449	0.572	1.030	0.955 1.111
Low vision	0.028	0.308	1.038	1.029	0.974 1.086
Syncope	0.000	0.999	0.000	1.000	0.819 1.221
**Vertigo**	**0.080**	**0.007**	**7.304**	**1.083**	**1.022 1.148**
Joint pain	0.040	0.116	2.470	1.041	0.990 1.094
Skin lesion	0.048	0.230	1.441	1.049	0.970 1.135
Sjogren’s syndrome	0.027	0.382	0.764	1.028	0.967 1.092
Raynaud’s phenomenon	0.094	0.195	1.683	1.099	0.953 1.267
Myalgia	0.043	0.094	2.801	1.043	0.993 1.097
**Dyspnea**	**0.057**	**0.054**	**3.699**	**1.059**	**0.999 1.122**
**Chest pain**	**0.057**	**0.029**	**4.790**	**1.059**	**1.006 1.115**
Sore throat	0.027	0.515	0.423	1.027	0.948 1.113
Sputum	0.013	0.785	0.075	1.013	0.922 1.113
Rhinitis	0.022	0.570	0.322	1.022	0.947 1.103
Lack of appetite	−0.003	0.950	0.004	0.997	0.917 1.085
Cog symptoms	0.012	0.752	0.100	1.012	0.942 1.087

Note: Results are corrected for the effect of age, presence/absence of hospital admission due to COVID-19, number of symptoms during SARS-CoV-2 infection, and time elapsed from COVID-19. Significant results are in bold. CI, confidence interval; OR, odds ratio.

## Data Availability

The data presented in this study are available on request from the corresponding author.

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
