# Peer review of "Mixed Depression in the Post-COVID-19 Syndrome: Correlation between Excitatory Symptoms in Depression and Physical Burden after COVID-19"

_brainsci, 2023, doi:10.3390/brainsci13040688_

Round 1

Reviewer 1 Report

Comments and Suggestions for Authors

This is an interesting paper focused on the investigation of excitatory symptoms in patients with depression in the post-COVID-19 syndrome. The paper is well written and of interest for the journal; however, several minor changes should be made.

ABSTRACT

1- I would start with: "Few studies have investigated the association between depression with excitatory symptoms and post-covid syndrome".

2-The main objective of the study cannot be to address "such lack". I recommend to define better the main aims of the study. What are the authors aiming to investigate?

3- The last sentence of the abstract should mainly be focused on future perspectives. I recommend to add future directions.

INTRODUCTION

1- I recommend to expand the paragraph explaining what a mixed depression is (lines 77-82).

2- The main aims of the study should be reported in a separate subsection (example: 1.1. Aims).

MATERIALS AND METHODS

1- The term "contracted" is a bit confusing. I prefer, who suffered from that infection.

2- Primary outcomes should be also clarified.

RESULTS

1- Sociodemographic characteristics and data regarding psychiatric evaluation should be described in a general section called "Main characteristics of the sample".

2- Women are underrepresented in the depression group. Is there any reason that the authors need to discuss?

Table 3 is repeating the term "current". I recommend to put them in the title and avoid to repeat it.

DISCUSSION

1- The discussion section should not be a summary. I recommend to explain the main findings at the beginning, and after that, explain each result and discuss with the literature. 

2- The limitations have been well documented. I recommend to expand the strenghts of the study. 

A section called future perspectives would be necessary. For instance, the authors can also hypothesize which kind of antidepressants o which pathways to be explored.

Reviewer 2 Report

Comments and Suggestions for Authors

This study aims to investigate the relationship between Post-Coronavirus-19 disease syndrome and mixed depression, a specific sub-form of depression with high levels of excitatory symptoms. A total of 1460 subjects with Post-COVID-19 syndrome were assessed, with those with depression showing higher percentages of family history of psychiatric disorders and post-COVID-19 symptoms. Higher levels of excitatory symptoms were associated with higher frequencies of post-COVID-19 syndrome symptoms. Treatment strategies to address depression may help ease the burden of post-COVID-19 syndrome.

The sample is adequate. It consisted of 1460 subjects who contracted SARS-Cov-2 infection and underwent a multidisciplinary evaluation in a Post-Acute Care Service at the Fondazione Policlinico Universitario Agostino Gemelli IRCCS of Rome, Italy. The study applied inclusion and exclusion criteria, and the variables considered included sociodemographic characteristics, psychiatric evaluation data, and multidisciplinary assessment data. The study was also approved by an ethical committee and informed consent was obtained from the subjects involved.

The sample was divided into two groups according to the presence/absence of depressive symptoms: 1) subjects with an HAM-D total score >7 were defined as subjects with depression (DEP). Subjects with a HAM-D score <7 were defined as comparisons (COMP). COMP also needed to have absence of unstabilized psychopathology, as confirmed by a Hamilton Anxiety Rating Scale (HAM-A) score below 17 and a Brief Psychiatric Rating Scale (BPRS) total score below 25. It is not clear why the cut-off is 7. Please elaborate in the text.

I would shorten a bit the discussion.

In general, the study is interesting

Comments on the Quality of English Language

it is very well written
